# SpitWorm, a Herbivorous Robot: Mechanical Leaf Wounding with Simultaneous Application of Salivary Components

**DOI:** 10.3390/plants8090318

**Published:** 2019-08-31

**Authors:** Guanjun Li, Stefan Bartram, Huijuan Guo, Axel Mithöfer, Maritta Kunert, Wilhelm Boland

**Affiliations:** 1Department of Bioorganic Chemistry, Max Planck Institute for Chemical Ecology, Hans-Knöll-Str. 8, D-07745 Jena, Germany; 2Department of Natural Product Biochemistry, Max Planck Institute for Chemical Ecology, Hans-Knöll-Str. 8, D-07745 Jena, Germany; 3Leibniz Institute for Natural Product Research and Infection Biology—Hans-Knöll-Institute (HKI), Beutenbergstr. 11a, D-07745 Jena, Germany; 4Research Group Plant Defense Physiology, Max Planck Institute for Chemical Ecology, Hans-Knöll-Str. 8, D-07745 Jena, Germany

**Keywords:** plant defense, herbivory, mechanical wounding, oral secretions, induced volatiles, spitworm, mecworm, *Phaseolus lunatus*, *Spodoptera littoralis*, volatile organic compounds

## Abstract

Induction of jasmonate-mediated plant defense against insect herbivory is initiated by a combination of both mechanical wounding and chemical factors. In order to study both effects independently on plant defense induction, SpitWorm, a computer-controlled device which mimics the damage pattern of feeding insect larvae on leaves and, in addition, can apply oral secretions (OS) or other solutions to the ‘biting site’ during ‘feeding,’ was developed and evaluated. The amount of OS left by a *Spodoptera littoralis* larva during feeding on *Phaseolus lunatus* (lima bean) leaves was estimated by combining larval foregut volume, biting rate, and quantification of a fluorescent dye injected into the larvae’s foregut prior to feeding. For providing OS amounts by SpitWorm equivalent to larval feeding, dilution and delivery rate were optimized. The effectiveness of SpitWorm was tested by comparing volatile organic compounds (VOC) emissions of *P. lunatus* leaves treated with either SpitWorm, MecWorm, or *S. littoralis* larvae. Identification and quantification of emitted VOCs revealed that SpitWorm induced a volatile bouquet that is qualitatively and quantitatively similar to herbivory. Additionally, RT-qPCR of four jasmonic acid responsive genes showed that SpitWorm, in contrast to MecWorm, induces the same regulation pattern as insect feeding. Thus, SpitWorm mimics insect herbivory almost identically to real larvae feeding.

## 1. Introduction

Standing at the beginning of the food chain, plants undergo biotic and abiotic challenges from the environment. In nature, herbivorous insects are one of their major threats, especially in vascular plants. Despite their physical immobility, plants have survived and propagated for hundreds of millions of years. During this long time, they have coevolved with herbivorous insects and developed strategies to fend, repel, and defeat their insect enemies [1]. Plant defense strategies against herbivores have aroused passionate and intense interests and research with profound achievements, especially in the last 30 years [2]. These studies have deciphered that the feeding of insects can initiate a series of diverse defense related events in planta, such as signaling processes, jasmonate accumulation, specific gene and protein expression patterns, and the production and accumulation of secondary metabolites, including volatile emissions [3].

But herbivory is more than the simple removal of plant tissue [4]. Besides the wounding trauma, defense responses of plants to an herbivore attack are triggered by compounds released by the herbivore. These stimuli are classified into two categories: (i) Chemical elicitors derived, for example, from herbivore oral secretions, oviposition fluids, or environmental DNAs (eDNA) that were left behind by insects (herbivore-induced molecular pattern, HAMPs); and (ii) those that originate from the specific patterns of wounding; i.e., the mechanical damage and the resulting elicitors from plants like oligosaccharides and peptides [5,6,7,8]. This second category is also called damage associated molecular patterns (DAMPs). Only both aspects together are able to induce the full spectrum of plant herbivory defenses [9].

To study the contributions of the two aspects (mechanical wounding and chemical elicitors) the insect’s feeding behavior has to be emulated and separated from the ‘insect’s chemistry’ [10]. Mechanical wounding of insect feeding was originally mimicked with different tools, including razor blades [11,12,13], pattern wheels [14,15,16], forceps [17,18,19], paper punches [20], and needles [21]. However, using the lima bean (*Phaseolus lunatus*) as a model plant, mechanical wounding alone by cuts or scratches did not induce volatile emission [22]. Only continuous mechanical wounding by a computer-controlled device (MecWorm) which mimics the leaf wounding pattern of a feeding insect, caused an intense emission of a blend of volatiles [9,22]. Those results indicated that mechanical wounding itself plays important roles in plant defense induction. Before the introduction of MecWorm, mechanical wounding with single or a few cuts or scratches with different wounding tools was used as a control or in combination with larval oral secretions (OS) or OS elicitors to study the defense inducing roles of chemical compounds from OS. This was effective in inducing plant defense responses, such as volatile emission and jasmonic acid (JA) burst [23].

These elicitors include low molecular weight fatty acid—amino acid conjugates (FACs) [23,24,25,26]; inceptins [27,28]; caeliferins [28] and volicitin [29]; glucose oxidase (GOX) [30] or a *β*-glucosidase [31]; and pore or channel forming compounds [32,33] reported to induce signaling pathways, biosynthesis of phytohormones, and volatile emissions. Compared with the vast diversity of herbivores that attack plants, the known herbivore-derived elicitors are relatively few. However, the molecular mechanism of plant perception of these known elicitors needs further study [34].

Methods to study insect OS or OS derived elicitors include mainly applying saliva or related components onto wounds to mimic insect feeding and examine plant defense response. Up to now, no standard procedure for wounding or OS application was established. Thus, besides different ways of wounding itself, varying OS amounts applied (1 to 20 μL), dilution factors up to 1:5, and differing wounding areas, ranging from a few scratches or puncture rows up to 2% of the total leaf, can be found in the various studies [14,16,35].

To examine the effect from different amounts and concentrations of insect OS applied to mechanical wounding, Musser, et al. [36] prevented the delivery of larval saliva (*Helicoverpa zea*) during feeding by cauterizing or surgically removing the larvae’s labial salivary glands. By using this technique, they showed that tobacco plant defense responses to caterpillar feeding were qualitatively different when caterpillars are either able or not able to secrete saliva. In another case, Major and Constabel [15] used a dilution range from 1:1 to 1:180 to optimize the aqueous dilution of OS from *Malacosoma disstria* applied to poplar leaves with over 100 puncture holes for maximal target gene induction (*PtdTI3*). The reported quantitative effects of insect saliva introduced to plant wounding area indicate that it is important to quantify the delivery ability of saliva from insect to plant; i.e., how much saliva is delivered per bite by insect.

However, from all the former studies, one can estimate that the quantities of OS applied to mechanical wounded plants were often several thousand times higher than the actual amount left behind at the wounding site by a larva per feeding bout, which was estimated in the range of 0.5 to 5 nL (*Heliothis virescens* feeding on corn and tomato plants, respectively) [37]. To precisely mimic insect feeding, it is necessary to determine the real amount left at the wound-zone by insect feeding.

Although the development of MecWorm provided deeper insight in our understanding of insect herbivory, it was necessary to take the next step in order to mimic insect feeding as closely as possible. Thus, here an insect feeding-mimicking device was engineered and established that combines both mechanical wounding and the simultaneous application of chemical elicitors to allow the study of their different influence on the plant’s wounding response. This so-called SpitWorm was tested in comparison with both MecWorm and *Spodoptera littoralis* larvae feeding on induced defense responses in lima beans.

## 2. Results

### 2.1. SpitWorm System Setup

SpitWorm, based on the robotic system MecWorm [22], was developed by adding a syringe connected to a capillary running through the inner-hollow of the ‘biting’ needle of MecWorm’s punch head up to a hole close to the needle tip. The syringe was actuated by a syringe pump to generate a stable and quantitative fluid delivery (Figure 1).

For headspace volatiles, collection of control and *S. littoralis* treated leaves, and for MecWorm and SpitWorm treatments, the un-detached test leaves were enclosed in an acryl glass case together with the punch head, and equipped with a closed-loop volatile collection pump system (Figure 2).

A schematic sketch of the several steps for determining the OS amount left on a leaf (V*b*) by *S. littoralis* can be found in the Appendix A. The values used in the model for calculating the volume of OS per bite (V*b*) left by a larva at the wounding edge (see Section 2.8) are indicated in the respective sub-chapters.

### 2.2. Wounding Sizes of Leaves Fed by S. littoralis Larvae

In order to adjust the wounding sizes (i.e., the amount of leaf area damaged or eaten) to be generated by MecWorm and SpitWorm, leaf wounding sizes of different larval feeding periods were measured. With four replicates for each treatment, the mean wounding sizes upon larval feeding were, after 5 min—0.30 ± 0.13 cm^2^, 1 h—0.93 ± 0.45 cm^2^, 3 h—1.81 ± 0.81 cm^2^, 9 h—5.49 ± 1.78 cm^2^, and 17 h—7.25 ± 1.02 cm^2^; see Figure 3a.

Feeding activities of untreated larvae (control) were compared with the feeding performance of larvae injected with different volumes of fluorescent dye solution into their foregut to determine the optimal injection volume for the subsequent experiments (Figure 3b). Injection volumes of 1 µL and 5 µL showed no significant differences (control mean, 29.98 ± 4.29 mm^2^; 1 µL mean, 29.88 ± 5.02 mm^2^; control ~ 1 µL, *p* = 0.999; 5 µL mean, 32.10 ± 5.54 mm^2^; control ~ 5 µL, *p* = 0.978) in leaf wounding sizes whereas injected volumes ≥ 10 µL led to a significant decrease in feeding activity (10 µL mean, 14.10 ± 4.03 mm^2^; 15 µL mean, 9.08 ± 9.26 mm^2^; control ~ 10 µL, *p* = 0.003; control ~ 15 µL, *p* = 0.0002).

Due to a stronger and clearer fluorescence signal at the wounding edges (Appendix A) an injection volume (V*i*; see Section 2.8) of 5 µL of Lucifer Yellow solution was used in subsequent experiments and together with a feeding time of 5 min (*t* = 300 s) in the model for determining the volume of OS left per bite (V*b*; see Section 2.8).

### 2.3. Residence Time of Fluorescent Dye in the Larval Foregut

Observation of fluorescent dye injected larvae under UV light showed that it takes 45 min to 1 h for the fluorescence dye to start moving through the whole body of insect to the anus (Figure 4). Therefore, all experiments with fluorescent dye and insect dissection were conducted immediately, or at least within 30 min after injection.

### 2.4. Estimation of Larval Foregut Volume

Based on the relatively simple structure of the foregut of a *S. littoralis* larva, its shape was taken as cylindrical. Measuring of dissected foreguts (*n* = 5) resulted in an average foregut length—l = 4.3 ± 0.8 mm; average width—d = 3.8 ± 0.4 mm, resulting in an average foregut volume (V*g*) of 49 ± 17.3 µL, (Appendix A) which was used in the model for calculating the amount of OS left per bite (V*b*; see Section 2.8).

### 2.5. Optimized Flow Rate for OS Delivery in SpitWorm

The optimized flow rate for fluid delivery in SpitWorm was evaluated by visually comparing ink trails left on a filter paper during moving the punch head on a zigzag path (Appendix A) and observing the needle tip of the punching head. As long as the delivery rate of the ink was sufficient to form a small droplet, a continuous trail of ink was obtained. At a delivery rate of 2.5 nL·s^−1^ the ink trace was weak and vanished after a short distance. With 5 nL·s^−1^ the trail got weaker during moving and vanished in the third line of the zigzag path. A delivery rate of 10 nL·s^−1^, which left an uninterrupted ink trail, was chosen for subsequent SpitWorm experiments.

### 2.6. Fluorescent Microscopy Images of Different Treatments

After treatment with a labeled larva (Figure 5a) and SpitWorm delivering labeled larval OS (Figure 5b), respectively, a distinct fluorescence signal could be detected at the wounding edges of the leaves. After treatment with MecWorm (Figure 5c) and cutting with a razor blade (Figure 5d) no fluorescence could be detected. The comparison of the OS trail left by larva and SpitWorm, showed that the insect’s OS went deeper into the vascular bundles of the leaf.

### 2.7. Fluorescent Dye Quantification of Tissues at the Wounding Edges after Different Treatments

In order to adjust the amount of OS delivered by SpitWorm, concentrations of Lucifer Yellow were measured after extraction of wounding edge tissues of leaves treated by fluorescent labeled larvae and compared with the concentrations in extracts of wounding edge tissues after treatments with SpitWorm, delivering different dilutions of labeled larval OS. As shown in Figure 6, clear differences of the extracted amounts of fluorescent dye between larval and SpitWorm treatment were observed for 1:5 (mean ± SD, 20.52 ± 4.45 nL·mL^−1^, *p* < 0.001) and 1:30 (mean ± SD, 3.45 ± 1.29 nL·mL^−1^, *p* = 0.032) dilutions of labeled OS, respectively, whereas a dilution of 1:10 (mean ± SD, 7.99 ± 0.40 nL·mL^−1^, *p* = 0.816) resulted in a concentration range close to the average concentration of OS (mean ± SD, 8.46 ± 0.52 nL·mL^−1^) left at the wounding edges by labeled larvae (C*d*; see Section 2.8).

### 2.8. Estimation of OS Amount Left by the Insect into Plant Wounds Per Bite

The observation of larvae that were feeding on lima bean leaves revealed a biting rate (BR) of 3–5 bite·s^−1^. Combining the BR with the mean foregut volume (V*g*, 49 µL), the amount of fluorescent dye solution injected (V*i*, 5 µL), the average concentration of OS left at the wounding edges by labeled larvae (C*d*, 8.46 nL·mL^−1^), the feeding time (*t*, 300 s), and the solvent volume used for extraction of tissues from the wounding edges (V*s*, 1 mL), the amount of OS left per bite (V*b*) was calculated according to the following equation:$$Vb=\frac{Cd\times Vs\times \frac{Vg}{Vi}}{t\times BR}$$

Taking the three different BR into account the following volumes of OS left at the wounding edges per bite were calculated: 3 bite·s^−1^, 92 pL·bite^−1^; 4 bite·s^−1^, 69 pL·bite^−1^; 5 bites·s^−1^, 55 pL·bite^−1^; mean ± SD, 72 ± 18.6 pL·bite^−1^.

### 2.9. Volatile Organic Compounds Released upon Different Treatments

After optimizing the SpitWorm parameters, the next step was to test its abilities to provoke ‘insect feeding-like defenses.’ In the headspace of lima bean leaves treated with *S. littoralis* larvae, MecWorm, and SpitWorm, 38 different compounds were identified and quantified relative to an internal standard (IS). Figure 7 shows the comparison of the relative amounts for each identified compound released upon the different treatments. Identified compounds, their retention indices, relative amounts, and significance levels (*p*-values) of a pairwise comparison are listed in Appendix A. One-way ANOVA followed by Tukey’s HSD as post hoc test for each compound, comparing all treatments, showed that between larval and SpitWorm treatments, the relative amounts of only four compounds out of 38 were significantly different. In contrast the mean values of 23 compounds differed significantly between larval and MecWorm treatment (Figure 7, Appendix A).

A principal component analysis (PCA) of all treatments and relative amounts of all 38 compounds revealed significant differences between MecWorm treatment and SpitWorm and *S. littoralis* larval treatment, whereas confidence areas (95%) of SpitWorm and *S. littoralis* treatments overlap almost completely (Figure 8). The two principal components, PC1 and PC2, explain 55.2% of all observed variances.

### 2.10. Comparative Quantitative Real-Time RT-qPCR

For all time periods, the four different JA responsive genes tested (lipoxygenase (*LOX3*), phenylalanine ammonialyase (*PAL*), β-1,3-glucanase (*PR2*), and chitinase (*PR3*)) showed no significant differences in expression levels between *S. littoralis* and SpitWorm treatment. In cases where larval treatment resulted in a significant difference compared to sole mechanical wounding by MecWorm, SpitWorm and larval treatment showed a stronger induction (Figure 9).

## 3. Discussion

Plants react to herbivory with a series of defense reactions provoked by the mechanical destruction of plant tissue in combination with chemical compounds left by the feeding organism. For lima bean leaves, it has been shown that sole continuous mechanical wounding by a designed mechanical caterpillar (MecWorm) is able to induce volatile emissions qualitatively almost identical to the bouquet released upon herbivory by *S. littoralis* larvae. This induction was not observed by sole wounding with razor blades or pattern wheels [9,22], or in combination of this kind of wounding with the application of OS or volicitin [38]. These results made it questionable if experiments using single scratches, puncturing, cutting, etc. with simultaneous application of OS or single elicitors in amounts far beyond the ‘true’ amounts of natural feeding (see introduction), are the appropriate way to study plant defense responses.

In order to be able to study the influence of larval oral secretions or single chemical compounds on the leaf’s wounding response, we aimed for turning the mechanical caterpillar MecWorm into SpitWorm, which combines the mimicking of mechanical leaf wounding by a larva with a continuous and simultaneous delivery of larval OS through a capillary to the tip of the punching needle. The feeding of *S. littoralis* larvae on lima bean leaves and the respective defense response of the plant was chosen as the ‘gold standard’ model for developing and parameter-adjusting of SpitWorm.

The average leaf wounding sizes of larval feeding were measured for different feeding periods (Figure 3a) and the continuous mechanical leaf damage by SpitWorm was set accordingly for experiments comparing larvae and SpitWorm treatment. For SpitWorm fluid delivery an optimized flow rate of 10 nL·s^−1^ was evaluated to leave a continuous trail during mechanical ‘biting’. It is worth noting, that due to the mechanical restrictions of an artificial device, the values for flow rate, as well as for biting rate and the leaf area destroyed per bite of SpitWorm, are different to real *S. littoralis* feeding (see below).

In order to determine the amount of OS left by a larva at the wounding edge and to adjust the amount of OS to be delivered by SpitWorm, a fluorescent dye solution (Lucifer Yellow CH) was injected into the foregut of the larvae before they started feeding on leaves. Lucifer Yellow CH was chosen because (i) of its fluorescence emission maximum at 535 nm, fitting perfectly in the green gap between 490 to 620 nm of chlorophyll a and b (no background at the wavelength of interest), (ii) it is assumed to be non-toxic, (iii) it has a high quantum yield, and (iv) it is highly dissociated at physiological pH levels [39]. A long-time effect of Lucifer Yellow on further larval growth and development was not evaluated in this study, because all experiments with fluorescence labeled larvae were completed within less than 1 h after injection.

Comparison of microscope fluorescence images of the wounding edges of leaves treated with labeled larvae, SpitWorm delivering fluorescent labeled OS (dilution. 1:10; delivery rate, 10 nL·s^−1^), and on the other hand, treated with MecWorm and with a razor blade, both without any fluorescent labeling, showed: (i) A pronounced fluorescence signal at the wounding edge after larva and SpitWorm treatment (Figure 5a,b), and (ii) no fluorescence after sole mechanical wounding by MecWorm or a razor blade (Figure 5c,d). This confirms that larval OS infiltrates the tissue at the wounding edge and that the fluorescence signal is not a resulting of, or affected by sole mechanical wounding. SpitWorm left a slightly different pattern of OS trail at the edge of the wounding site compared to larva feeding. Upon larval feeding, the OS goes apparently deeper into the plant tissue following the veins compared to SpitWorm treatment. The wounding edges showed a difference in biting patterns. In accordance with scanning electron micrographs of wounding sites resulting from MecWorm and larva treatment reported earlier [22], larva feeding forms a straight borderline in contrast to SpitWorm treatment, which forms to some extend a small, frayed zone which can explain the different permeation depths of the fluids.

The amount of OS left by a larva per bite (approximately 50 to 100 pL·bite^−1^) was calculated by quantifying the fluorescent dye extracted from the wounding edge tissues of leaves treated with labeled larvae in combination with feeding duration, average foregut volume, amount of fluorescent dye solution injected, and observed larval biting rates (3–5 bites·s^−1^). This resulted in a calculated flow rate of 250–300 pL·s^−1^ for the OS delivered by a larva, which is 30–40 times lower than the optimized flow rate (10 nL·s^−1^) determined for SpitWorm (see above). Thus, the ‘effective’ OS delivered by SpitWorm had to be aligned to the amount of OS from the larva during feeding by dilution. The amount of fluorescent dye extracted from larva-damaged tissue (see above) was compared with the amounts extracted from wounding edge tissues after SpitWorm treatment delivering different dilutions of labeled larval OS labeled with Lucifer Yellow solution (Figure 6). This resulted in an optimized dilution of 1:10 of larval regurgitate for SpitWorm treatments. The concentration of fluorescent dye in the regurgitate before dilution was adjusted to the concentration in the larval foregut.

With an OS dilution of 1:10 SpitWorm delivers an equivalent of 1 nL OS per second, which is still about three to four times the amount of OS left by larva feeding (0.3 nL·s^−1^). Here it needs to be considered that the feeding track of *S. littoralis* larvae is not linear but usually follows repetitively a curved path. The OS left in the tissue during a feeding bout is re-ingested by the larva in the next round [40], while SpitWorm is continuously delivering diluted OS without taking up the damaged plant tissue. Additionally, to reduce the viscosity caused by large polysaccharides, proteins, fat, and food residues, freshly harvested regurgitate was filtered through a 0.22 μm filter, and it cannot be excluded that active components were removed thereby.

Delivery of OS by SpitWorm was conducted at room temperature. Although the larvae were raised healthily within the same temperature range, it is not ensured that all the active compounds in the OS can keep the same activity, especially in experiments which last several hours. This problem may be compensated by an over-delivery of OS; nevertheless, more experiments need to be done in the future to test the activities of chemical factors by using SpitWorm.

In order to evaluate to what extent SpitWorm can mimic herbivory, relative volatile organic compound (VOC) amounts in the headspace of leaves, as well as expression levels of four JA responsive genes in leaves upon larvae, MecWorm, and SpitWorm treatment, were compared.

Volatiles in the headspace of lima bean leaves released upon different treatments were collected, identified, and quantified relative to an internal standard by GC-MS (Appendix A). Instead of charcoal as described earlier for comparison of MecWorm and larvae treatment [9,22], we used Porapak Q as adsorbent for a more reliable quantification. It is known that, for example (*E*)-β-ocimene, one of the most abundant compounds induced after larvae treatment of lima bean leaves, is oxidized to (3*E*,5*E*)-2,6-dimethyl-3,5,7-octatrien-2-ol and dehydrogenated to (3*E*,5*E*)-2,6-dimethyl-1,3,5,7-octatetraene to some extent when collected on active charcoal in the presence of humid air [41]. Besides those two, a third artefact—5-ethylfuran-2(5H)-one (described as hexenolide by Bricchi, et al. [9]), only detected after continuous mechanical wounding by MecWorm, but not upon larval treatment—was absent when using Porapak Q as adsorbent. In total 38 compounds in all three treatments were identified and their relative amounts were subjected to a principal component analysis, which revealed an almost complete overlap of the confidence areas (confidence level, 95%) resulting from larvae and SpitWorm treatment. The cluster of the MecWorm treatments is clearly separated (Figure 8), showing that SpitWorm mimics larval feeding much better than MecWorm. Comparing the relative amounts of each compound after SpitWorm and larvae treatment shows that 90% are not significantly different. On the other hand, comparing MecWorm with larval treatment, the relative amounts for 60% of the compounds exhibit a significant difference (Appendix A). In general, MecWorm treatment evoked a stronger plant response, i.e., higher amounts of compounds released, than larvae or SpitWorm treatment (Figure 7, Appendix A). This leads to the conclusion, that in this case, mechanical wounding is responsible for inducing volatile emission upon herbivory, figuratively named ‘the cry for help,’ but compounds in the OS reduce the emission to ‘turn down the sound’.

To further investigate responses to SpitWorm treatment in *P. lunatus* leaves, time series of expression levels for four JA-responsive genes were chosen, which were also used in earlier studies [42]. The four genes encode for: Lipoxygenase (*LOX3*), phenylalanine ammonialyase (*PAL*), and the pathogenesis-related (PR) proteins (*PR2* (β-1,3-glucanase)) and (*PR3* (chitinase)). All treatments showed no significant difference between SpitWorm treatment and larvae feeding (Figure 9). Whereas for *PAL* SpitWorm/*S. littoralis*, treatments resulted in a stronger induction for all periods compared to sole continuous mechanical wounding by MecWorm (Figure 9), the expression levels of the other JA responsive genes showed no significant differences between the different treatments for longer periods. Except for *LOX* where almost no difference between mechanical wounding and larval feeding is observed, the early response (1 h) was influenced by OS from *S. littoralis* or SpitWorm, respectively. This indicates that mechanical wounding alone is able to induce the JA responsive genes pathway, but chemical factors enhance or modulate this induction for a more rapid defense response. Additionally, the results showed that 1:10 dilution of OS is an appropriate dilution factor to add OS to SpitWorm to mimic *S. littoralis* feeding.

As a further developed MecWorm, a SpitWorm was expected to mimic the action of a feeding insect as close as possible. The results emphasize that both mechanical wounding and chemical factors play prominent roles in gene regulation and defense reactions, which further proves that SpitWorm can be used as an effective tool in mimicking insect feeding. Our findings also support the hypothesis that in wounded leaves, mechanical wounding can trigger most of the defense reactions, while chemical factors in insect OS have a ‘fine-tune’ function by enhancing or attenuating the induction of gene expression by mechanical wounding.

With this new tool at hand, it is now possible to study the interplay of mechanical wounding and larval OS at different environmental conditions or with different combinations of compounds. Using fractions of larval OS or single compounds will allow tracking down individual elicitors, and in combination with other comparative genomic, transcriptomic, or proteomic methods, it will be possible to go further and deeper in understanding regulation processes of plant defense against herbivory.

## 4. Materials and Methods

### 4.1. Plant and Insect Materials

#### 4.1.1. Plants

Lima beans, *Phaseolus lunatus* L. (Ferry Morse cv. Jackson Wonder Bush) were grown from seeds at 23 °C and 60% humidity in plastic pots (diameter 5.5 cm) using sterilized potting soil. For daylight radiation, fluorescent tubes (ca. 270 μE m^−2^ s^−1^) with a photophase of 16 h were used. Experiments were conducted with 12 to 16 day old seedlings, showing two fully developed primary leaves.

#### 4.1.2. Insects

*Spodoptera littoralis* (Lepidoptera, Noctuidae) larvae, hatched from eggs (Bayer CropScience AG, Monheim, Germany) were reared on artificial diet (500 g white beans powder soaked overnight in 1.2 L water, 9 g ascorbic acid, 9 g parabene, 4 mL formaldehyde (36.5%), and 75 g agar boiled in 1.0 L of H_2_O) and raised at 22 °C to 24 °C, with a 14 h to 16 h photophase, to the developmental stage of 3rd to 5th instar. For all experiments, larvae with a body length in the range of 2.5 to 3 cm were chosen.

### 4.2. General Conditions

For all plant treatments (*S. littoralis*, MecWorm, SpitWorm), one un-detached primary leaf of a seedling was inserted in the cubicle of the robotic device. Temperature, humidity (not controlled), and illumination regime were identical to the larvae raising conditions. Three MecWorms/SpitWorms were used simultaneously, each with a punching unit consisting of a hollow needle with 0.5 mm diameter. The punching strokes were adjusted to destroy squared areas of leaf material according to the respective mean wounding sizes and a similar pattern resulting from larval herbivory. A damage of the leaf’s main rib was never observed on larval feeding. Therefore, the wounding pattern of the mechanical devices was set accordingly.

### 4.3. SpitWorm

A gas-tight glass syringe (100 mL) was connected to a capillary (Fused Silica, 0.25 mm i.d., SGE, Melbourne, Australia) running from the top of the punching head of MecWorm [22] through the inner-hollow of the ‘biting’ needle up to to a hole close to the needle tip. The syringe was actuated by a syringe pump (Harvard Apparatus PHD 2000) to generate a stable and quantitative fluid delivery (Figure 1).

### 4.4. Flow Rate Optimization

In order to determine the lowest flow rate at which a fluid can be supplied continuously without interruption through the hole in the punching needle, ink was used instead of larval OS. With different delivery rates (2.5 nL·s^−1^, 5 nL·s^−1^, and 10 nL·s^−1^) SpitWorm was set to mimic larval biting pattern on a filter paper for 5 min.

### 4.5. Collection of Insect Oral Secretions

Regurgitate was collected by slight squeezing the larvae with tweezers and collection of the excreted fluid with a Gilson Pipetman P20 Variable Volume Pipette (2 to 20 µL). If not used immediately, the regurgitate was frozen and stored at −20 °C. Before dilution and use with SpitWorm, larval regurgitate was filtered through a syringe filter (CME, 0.22 µm).

### 4.6. Wounding Size and Biting Rate Determination

After 12 h of starving, single larvae were allowed to feed on a lima bean leaf for 5 min, 1 h, 3 h, 9 h, and 17 h. For subsequent quantification of wounding areas, only treatments where the larva did not feed at the leaf edges were used. Images of the damaged leaves were printed out. Scaled unit areas and wounding areas were cut out of the same sheet of paper and weighed on an analytical scale. Wounding sizes were determined by dividing the paperweight of the wounding area by the paperweight of the scaled unit (*n* = 4 for each treatment).

Biting rate (bite·s^−1^; BR; see Section 2.8) was determined by evaluating a close-up slow-motion video of a larva during feeding on a lima bean leaf.

### 4.7. Insect Foregut Volume Determination

After feeding, larvae were euthanized by immersion in 75% ethanol solution for 30 s, dissected, and the length (l) and width (d) of the foregut were measured. Foregut volume (V*g*) was calculated as a cylinder (V*g* = l·π·(d/2)^2^, *n* = 5).

### 4.8. Optimization of the Injection Volume of Fluorescent Dye Solution

After being starved for 12 h, larvae were injected with 1 µL, 5 µL, 10 µL, and 15 µL of an aqueous solution (1 mg·mL^−1^) of Lucifer Yellow CH dipotassium salt (Fluka, *λ_Ex_* = 428 nm, *λ_Em_* = 535 nm), with 4 replicates for each injection volume. As controls, 4 starved larvae, not-injected, were used. Larvae were then fed on lima bean plants for 5 min, wounding areas were pictured with a LEICA LMD6000 fluorescence microscope, and wounding sizes were measured as described above.

### 4.9. Fate of Foregut Injected Fluorescent Dye

In order to estimate the residence time of the injected fluorescent dye solution in the larval foregut, *S. littoralis* were injected with 5 µL of an aqueous solution (1 mg·mL^−1^) of Lucifer Yellow and observed under ultraviolet light (365 nm) for 3 h.

### 4.10. Amount of OS Left on the Leaf at the Wounding Zone

An aqueous solution (5 µL, 1 mg·mL^−1^) of Lucifer Yellow CH dipotassium salt was injected into the larval foregut. After injection, larvae were allowed to feed on leaves for 5 min. Leaf tissue around the wound margins was cut out, ground in liquid nitrogen, suspended in 1 mL H_2_O by shaking for 1 h in the dark at 4 °C, and centrifuged for 10 min at 12.6 × 1000 rcf. The supernatant was subjected to fluorescence signal quantification with a FP-750 Spectrofluorometer (JASCO International Co. Ltd., Tokyo, Japan). The supernatant of centrifuged tissue suspensions of leaves treated with unlabeled larvae, processed in the same manner as above, served as blank and as the solvent for the dilutions for the standard curve measurements (*n* = 3 for each treatment). A standard curve (R^2^ = 0.9968; Appendix A) was generated using a series of dilutions (0, 5, 10, 15, 20, 25, and 30 nL·mL^−1^) of Lucifer Yellow solution (1 mg·mL^−1^).

### 4.11. Optimization of OS Dilution for SpitWorm Treatment

Leaves were treated for 5 min with SpitWorm (wounding size, 0.30 cm^2^; fluid delivery rate, 10 nL·s^−1^) using different aqueous dilutions (1:5, 1:10, and 1:30) of fluorescence dye labeled regurgitate (Lucifer Yellow solution (5 µL, 1 mg·mL^−1^) in 44 μL of filtered larval regurgitate).

For fluorescence signal quantification (*n* = 3 for each treatment), tissues of the wound edges were processed in the same manner as described for the treatment with labeled larvae (see above).

### 4.12. Fluorescence Microscope Imaging

Lima bean leaves were treated by *S. littoralis* larva (injected with Lucifer Yellow solution, 5 µL, 1 mg·mL^−1^), SpitWorm (delivering a diluted (1:10) Lucifer Yellow solution (5 µL, 1 mg·mL^−1^) in 44 μL of filtered larval regurgitate at a flow rate of 10 nL·s^−1^), and by MecWorm and a razor blade, both without fluorescent labeling. Pictures of the wounding areas were taken with a LEICA LMD6000 fluorescence microscope.

### 4.13. Collection and Analysis of Headspace Volatiles

For headspace volatiles collection of *S. littoralis* treated (one larva per leaf), as well as for MecWorm and SpitWorm (filtered larval OS, 1:10 diluted; flow rate, 10 nL·s^−1^) treated leaves (equivalent duration (17 h) and wounding areas (7.25 cm^2^) for all treatments), the test leaves were enclosed in an acryl glass case (width × depth × height, 95 × 87 × 135 mm^3^; net headspace volume approximately 1 L) together with the punch head in the MecWorm device (Figure 2). The capillary of SpitWorm was threaded through a hole (diameter, 0.5 mm) in the left sidewall of the case; the stainless steel tubing of the volatile collection pump system was inserted through two holes (diameter, 0.8 mm) in the top-side. For control (untreated leaves) and larvae treatment, the punching head was positioned away from the leaf area and the device was switched off. The wounding time and area were 17 h and 7.25 cm^2^, respectively. Headspace volatiles emitted by untreated lima bean leaves (*n* = 8), or treated with larvae (*n* = 6), MecWorm (*n* = 5), and SpitWorm (*n* = 6), were continuously collected for 24 h on Porapak Q traps (quartz glass tubes; length, 66 mm; inner diameter, 2.5 mm; outer diameter, 5.5 mm; filled with 10 mg Porapak Q, 80–100 mesh, Aldrich) using the closed-loop-stripping (CLS) method [43]. Traps were pre-cleaned before the first use and regenerated after elution by rinsing with 1 mL of solvent each in the following order: Methanol, methanol/chloroform (1:3), chloroform, acetone, dichloromethane, and dried at 60 °C for 24 h. All experiments were started between 11:00 and 13:00. Setups were kept at 22–24 °C with a light/dark rhythm of 7 h light, 10 h dark, and 7 h light. For all samples after volatile collection, adsorbed compounds were eluted with dichloromethane (2 × 50 μL, supplemented with 1–bromodecan (50 µg·mL^−1^) as internal standards, and stored at −20 °C prior to analysis. Samples were analyzed by gas chromatography mass spectrometry (GC-MS) with an ISQ GC-quadrupole MS system (Thermo Scientific, Bremen, Germany) equipped with a fused silica capillary column ZB–5 (30 m × 0.25 mm × 0.25 µm with 10 m guard column, Zebron, Phenomenex, CA, USA). Helium at 1 mL·min^−1^ served as carrier gas with an injector temperature of 220 °C running in split mode (1:10); 1 µL of sample was injected. Separation of the compounds was achieved under programmed temperature conditions (45 °C for 2 min, then at 10 °C·min^−1^ to 200 °C, then at 30 °C·min^−1^ to 280 °C kept for 1 min). The MS was run in EI mode (70 eV) with a scan range of 35 to 450 amu, a transfer line temperature of 280 °C, and an ion source temperature of 250 °C. Data acquisition was performed using Xcalibur 3.1 (Thermo Fisher Scientific).

A mixture of n-alkanes C_8_–C_20_ in *n*-hexane (Aldrich) was measured before and after a sample sequence under the same conditions except for the injector split ratio (1:50). Retention times of the *n*-alkanes were used to calculate the retention index (RI) for each peak in the GC-MS chromatogram according to the method of van Den Dool and Kratz [44].

Compounds were identified based on their mass spectra (MS) in combination with their individual RIs in comparison to NIST [45], Adams [46] and Massfinder [47] MS and RI databases, using Massfinder software in combination with NIST MS Search. Authentic reference compounds were used additionally for identification, if at hand. For relative quantification, identified peaks of the GC-MS total ion chromatogram (TIC) were integrated and the peak areas were divided by the peak area of the internal standard.

### 4.14. One-Step Comparative RT-qPCR

For gene expression analysis, lima bean leaves were treated by larvae, MecWorm, and SpitWorm (flow rate, 10 nL·s^−1^; OS dilution, 1:10) for 1, 3, and 9 h with comparable leaf wounding sizes for each period. Leaves of untreated plants served as control. Three technical and three biological replicates were used for each sample.

Primers for RT-qPCR were designed with Primer3plus [48] according to gene sequences from *Phaseolus vulgaris*: Lipoxygenase (*LOX3*, X63521, [49]) in the octadecanoid pathway; phenylalanine ammonialyase (*PAL*, M11939, [50]) in the phenylpropanoid pathway; and pathogen-related (PR) proteins (*PR2* (β-1,3-glucanase), X53129, [51]) and *PR3* (chitinase), M13968 [42,51,52]. *P. lunatus* actin (*PACT1*, DQ159907) gene was used as the normalizer [53]. OligoAnalyzer 3.1 was used for primer analysis.

Primers used for RT-qPCR were as follows were follows:

*LOX3* (X63521),forward5′-TGGATGACCGATGAAGAA-3′reverse5′-TGTTGCTATGACGAATGG-3′*PAL* (M11939),forward5′-GAAACCTTAGAATCCATCACCA-3′reverse5′-TAGAAGCCAAGCCAGAACC-3′*PR2* (X53129),forward5′-AAACTCCTACCCTCCATCACAA-3′reverse5′-CCATCCCTCACCACAACA-3′*PR3* (M13968),forward5′-GGCGAGGACAGGATAGCAG-3′reverse5′-TCACAAAGGGAAACACAGATT-3′*PACT1* (DQ159907),forward5′-AGGCTCCTCTTAACCCCAAG-3′reverse5′-GTGGGAGAGCATAACCCTCA-3′

Leaf tissues (80–100 mg) were collected around the leaf’s wounding edges or from intact leaves as a control and ground in liquid nitrogen. Total RNA was isolated using Trizol reagent (Invitrogen) following the manufacturer’s protocol and purified using TURBO^TM^ DNase (Ambion) and RNeasy MinElute Cleanup kit. RNA was then directly subjected to 1-step comparative quantitative real-time RT-qPCR using the Verso^TM^ SYBR Green 1-Step QRT-PCR Low ROX Kit (ABgene), with an Mx3000P Real-Time PCR system (Stratagene). The process was conducted according to the manufacturer’s protocol with a 25 µL reaction system, consisting of 0.25 μL Verso Enzyme Mix, 12.5 μL 1-Step qPCR SYBR Mix, 1.25 μL RT Enhancer, 1.75 μL forward and reverse primers (1 μM) each, 2 μL RNA template (25 ng·μL^−1^), and 5.5 μL water (PCR grade). The reaction procedure was as follows: cDNA synthesis for 15 min at 50 °C, Thermo-Start activation for 15 min at 95 °C followed by 40 cycles of denaturation (15 s at 95 °C), annealing (30 s at 55 °C for *PR2* and *PAL*; 30 s at 60 °C for *LOX3* and *PR3*), and extension (30 s at 72 °C). Fluorescence signals were recorded after each annealing step. After this an additional temperature cycle (95 °C, 30 s and 60 °C, 30 s) was followed by an incremental heating to 95 °C (stepwise 0.5 °C for 10 s) to verify the products by a dissociation curve. Fluorescence signals were recorded during the whole melting process. PCR conditions were determined by comparing threshold values, followed by non-template control for each primer pair. Relative RNA levels were normalized with the level of *P. lunatus* actin mRNA (*PACT1*) and calibrated with relative expression levels of the target genes in untreated control plants.

### 4.15. Statistical Analysis

Statistical analysis was performed using the free software package R version 3.4.3 [54] in combination with RStudio version 1.1.423 [55]. Differences among group means for the various response variables were evaluated by one-way analysis of variance (one-way ANOVA), followed by post-hoc tests (Tukey’s HSD test or Fisher’s LSD) for multiple comparisons. RT-qPCR data were log2 transformed and Fisher’s LSD was used as post-hoc test.

For comparing the relative amounts of volatile compounds released upon different treatments, a dimension reduction by principal component analysis (PCA) with scaled experimental values was performed.

## Figures and Tables

**Figure 1 plants-08-00318-f001:**
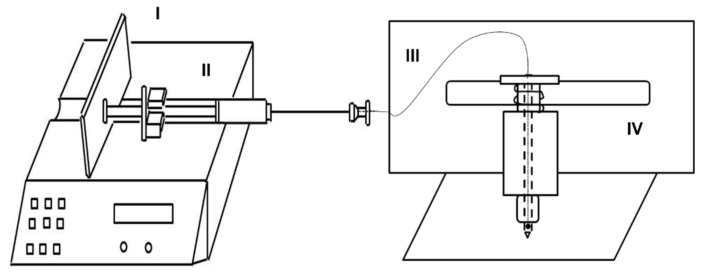
Schematic sketch of SpitWorm. (I) Syringe pump controlling the fluid flow rate; (II) syringe (100 μL) filled with diluted oral secretions (OS); (III) capillary from the syringe to the tip of MecWorm’s hollow punching needle; (IV) MecWorm, a robotic system for continuous wounding.

**Figure 2 plants-08-00318-f002:**
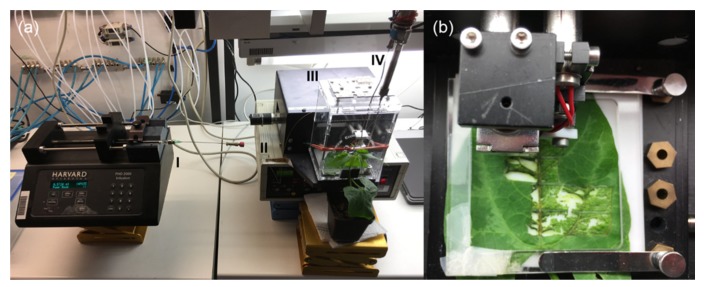
Volatile collection with SpitWorm. (**a**) MecWorm/SpitWorm. (I) Automatic syringe pump with a 100 μL syringe; (II) capillary for OS delivery; (III) plexiglas case; (IV) volatile collector. (**b**) Close-up of the biting head of SpitWorm punching a lima bean leaf. For MecWorm treatment, the capillary was removed and for larval treatment, the system was switched off.

**Figure 3 plants-08-00318-f003:**
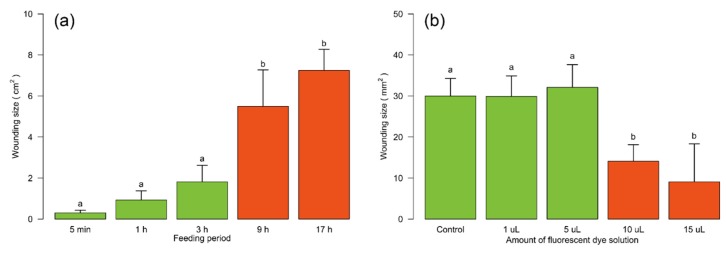
Wounding sizes of lima bean leaves. (**a**) *Spodoptera littoralis* feeding for 5 min, 1 h, 3 h, 9 h, and 17 h. (**b**) After 5 min feeding of *S. littorals*, injected with different volumes of fluorescence dye solution. Larvae not injected served as controls. Mean ± SD; *n* = 4; one-way ANOVA; post hoc test: Tukey’s HSD; treatments with identical letters are not significantly different.

**Figure 4 plants-08-00318-f004:**
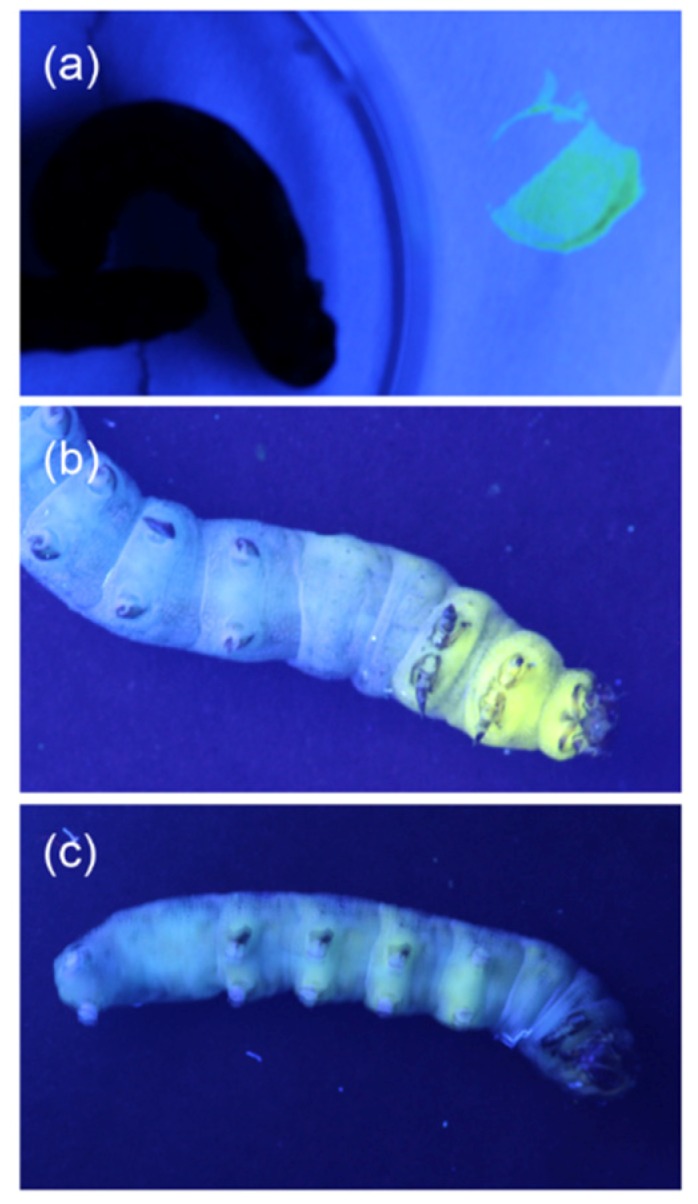
Fluorescence images of an *S. littoralis* larva. (**a**) Larva before injection (on the right: A spot of Lucifer Yellow on filter paper), (**b**) 10 min, and (**c**) 1.5 h after injection of Lucifer Yellow solution (5 µL, 1 mg·mL^−1^) into the larval foregut.

**Figure 5 plants-08-00318-f005:**
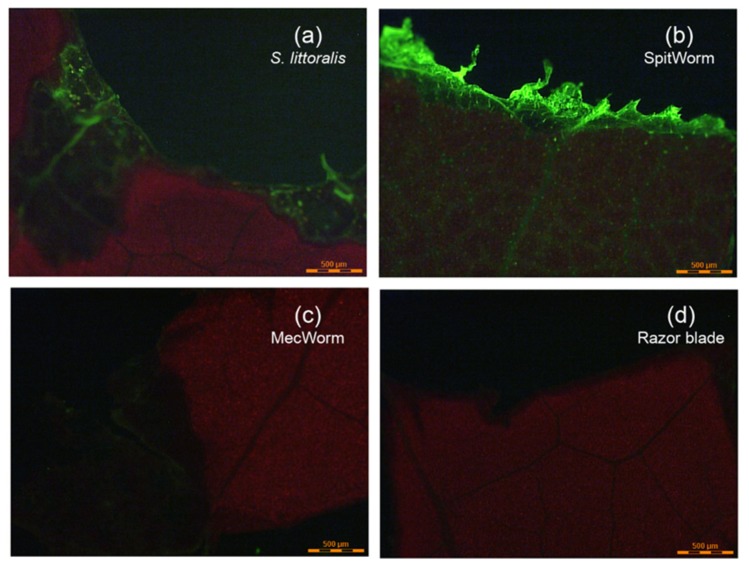
Fluorescent microscopy images of lima bean leaves after different treatments. Detail view of wounding edges of the adaxial leaf surface. Treatments: (**a**) *S. littoralis* larva injected with Lucifer Yellow solution; (**b**) SpitWorm delivering Lucifer Yellow labeled larval OS; (**c**) MecWorm; and (**d**) cut with a razor blade.

**Figure 6 plants-08-00318-f006:**
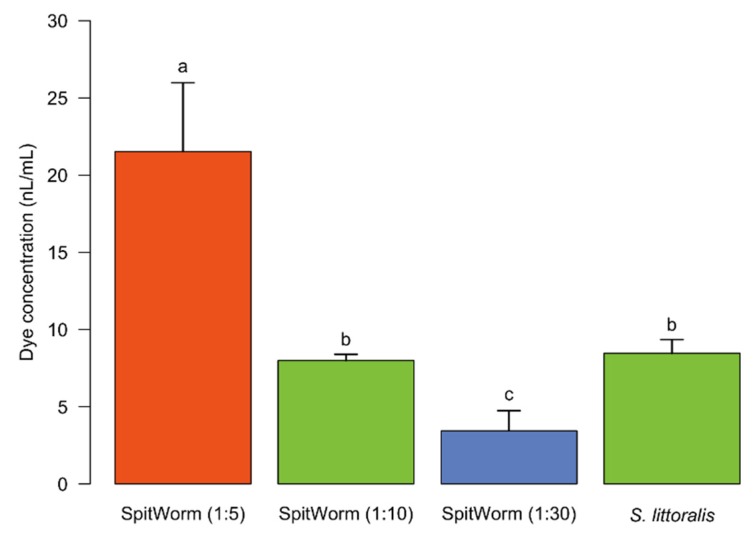
Fluorescent dye concentrations extracted from wounding edge tissues after different treatments. Treatments: *S. littoralis* injected with fluorescent solution; SpitWorm delivering different dilutions of labeled larval OS (1:5, 1:10, 1:30); *n* = 3 for each treatment; mean ± SD; one-way ANOVA; post hoc test: Fisher’s LSD; treatments with identical letters are not significantly different.

**Figure 7 plants-08-00318-f007:**
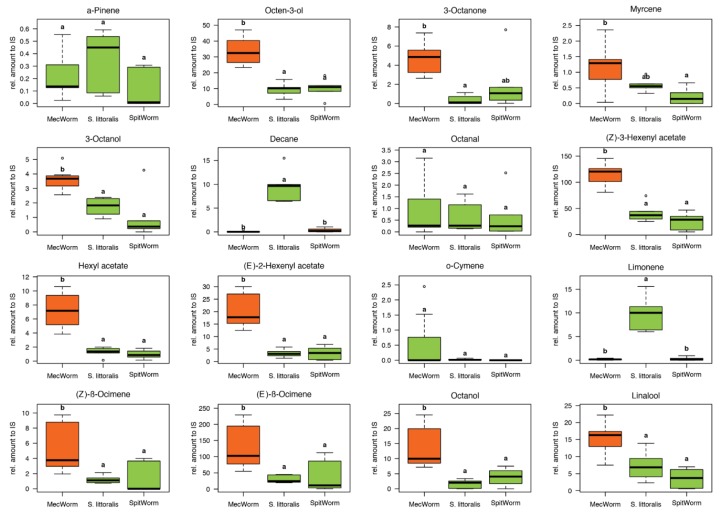
Comparison of relative amounts of headspace volatiles upon different treatments. Three different treatments on lima bean leaves (*S. littoralis* larvae, *n* = 6; MecWorm, *n* = 7; SpitWorm, *n* = 6). One-way ANOVA; post hoc test: Tukey’s HSD; treatments with identical letters showed no significant difference; equal colors indicate no significant difference to SpitWorm treatment.

**Figure 8 plants-08-00318-f008:**
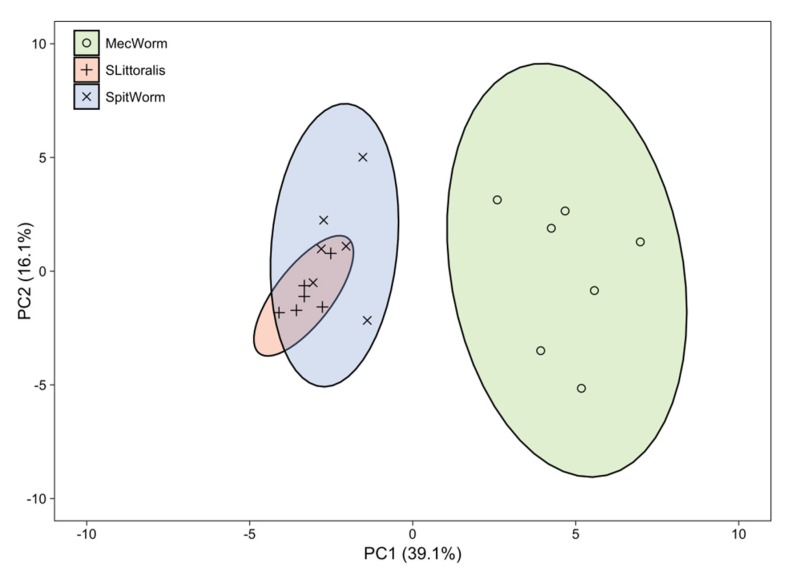
Principal component analysis of relative amounts of 38 volatiles released by different treatments. Three different treatments on lima bean leaves (*S. littoralis* larva, *n* = 6; MecWorm, *n* = 7; SpitWorm, *n* = 6). PC, principal component (% of total variance); confidence area, 95%.

**Figure 9 plants-08-00318-f009:**
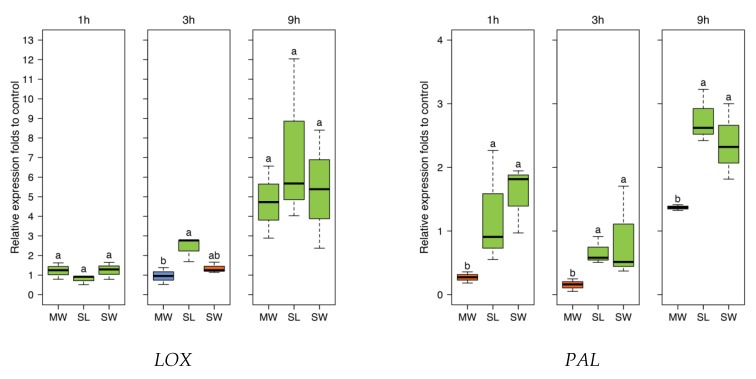
Expression of four JA responsive genes (*LOX3*, *PAL*, *PR2*, and *PR3*). Lima beans treated for 1 h, 3 h, and 9 h with MecWorm (MW), *S. littoralis* (SL), and SpitWorm (SW). *Phaseolus lunatus’s* actin housekeeping gene (*PACT1*) served as normalizer. (SW; 1:10 diluted OS, delivery speed of 10 nL·s^−1^); *n* = 3 for each treatment; log2 transformed; one-way ANOVA; post-hoc test: Fisher’s LSD; treatments with identical letters are not significantly different.

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
