# Peer review of "SpitWorm, a Herbivorous Robot: Mechanical Leaf Wounding with Simultaneous Application of Salivary Components"

_plants, 2019, doi:10.3390/plants8090318_

Round 1
Reviewer 1 Report
The paper describes an updated and optimized improvement on the MecWorm. I can see how it would be of value to researchers interested in untangling wound- and elicitor-induced plant responses, particularly with respect to identifying individual components of oral secretions that may modify host defenses.
The paper's experiments are evidently carefully designed and performed, and the authors very carefully word the results and discussion to assure adequate understanding of caveats to the conclusions.
I only see a few typographical oddities that disrupt the smoothness of reading:
lines 18 and 342: the commas after "...both" are unexpected
line 57: the 'd' after "induce" is not appropriate. The first "by" is not needed and the sentence can be simplified to "Only continuous mechanical wounding by..."
line 67: The "were" is not needed
line 198: There should be a hyphen in "...feeding-like defenses."
Author Response
Reviewer 1
lines 18 and 342: the commas after "...both" are unexpected
corrected
line 57: the 'd' after "induce" is not appropriate. The first "by" is not needed and the sentence can be simplified to "Only continuous mechanical wounding by..."
done
line 67: The "were" is not needed done line 198: There should be a hyphen in "...feeding-like defenses."
done
Reviewer 2 Report
The manuscript by Li et al. describes a device (called SpitWorm) which was developed for splitting the effects of herbivore oral secretions (OS) and mechanical wounding on plant responses to herbivory in a controlled experiment. This topic is of general interest, and many researchers will greatly appreciate the opportunities offered by this device. The manuscript describes new method in experimental botany and is therefore within the scope of 'Plants'. However, to my opinion, the manuscript requires substantial revision before it can be published.
'Discussion' is the weakest point of the manuscript. In the current version, it only duplicates the methods and results, so that discussion as such is almost lacking. I would expect here a comparison of the new system with other methods that have been used to simulate herbivory (listed in lines 54-55), with careful discussion of problems and benefits of all these methods. Another interesting point is to discuss the differences in relative amounts of individual volatile compounds induced by different methods – to which biosynthetic pathways these compounds belong and what is their (presumed) function in plant defences against herbivory? It would be also important to know whether the SpitWorm only exists in the lab of their creators, or there is a plan to make it (commercially?) available to other groups of researchers. Replication of a device is critical to justify the validity of the results.
Introduction is generally fine, although the justification of the need to control both mechanical wounding and OS application can be strengthened by adding references to meta-analyses which compared plant responses to natural and simulated herbivory. Also the goal of the study is missing in the introduction; it appears only in lines 237-239.
The starting part of the results (lines 98-117), to my opinion, should be moved to methods. The sentence «Results of … below» (lines 114-115) does not contain any information and can be deleted.
At several occasions (lines 120-122, 126-129, 176-179) the authors report numerical values both in the text and on figures. This duplication should be removed. Also Figure S5 is not needed, because it duplicates Table 1.
I am confused by the comparisons between wounding sizes of lima bean leaves resulting from larval feeding periods of different length (5 min to 17 hours; Figure 3a). I fully understand that the absolute values of the consumed leaf area are needed for calibrating SpitWorm; but for which purpose the authors compare between these areas? The only meaningful comparison between damage imposed during time periods of different length would be the wounding rate (mm2/hour).
The statement in line 164 («insect's OS went deeper…») is not supported by the data.
Fig. 5 is not really impressive, especially the panels (c) and (d), and difficult to understand and interpret. I suggest moving it to Supplementary materials.
The biting rate (line 186) is reported as an interval (minimum-maximum?), whereas all other data are reported as mean + SD. Was this measurement replicated?
Table 1 requires technical editing. The sign +/- before SD in the caption line should be removed. P values should be rounded to include 2 digits for p=0.10-0.99 and 4 digits for p=0.0001-0.0999. The three last columns are not needed, because actual p values are reported in the previous columns. If the authors wish, they may use e.g. boldface to highlight significant p values. Sample sizes (now shown at the bottom of the table) shall go to a caption; the numbers of components that show significant differences between treatments shall go to the text.
Materials and methods. I missed the following information: (1) Was the damage applied to one leaf per plant or to several leaves? (2) Were leaf size and leaf position controlled? (3) How the mechanical damage was applied – was it just punching by a needle? What was a needle diameter, what was a rate of damage? How the leaf was fixed during the period of damage? What was the shape of damage? (hole size, holes per leaf, hole position, etc.) (4) Were the environmental conditions (air temperature, illumination, humidity etc.) standard during all the experiments? These values need to be reported (as it is done for plant growth and insect rearing conditions), because e.g. insect feeding rate naturally depends on temperature. (4) Was there only one specimen of SpitWorm, which means that the replicated damage was applied one after another, or there were several devices so that the replicated damage was applied simultaneously?
The language of the manuscript is clear, but some expressions used by the authors (e.g. «annihilate their insect enemies», line 40; «insect feeding like defenses», line 198) seems odd for me. The authors used relevant literature, although several points in the beginning of Introduction are poorly supported by references – the cited papers are too narrow to support the very general statements.
Author Response
Reviewer 2
'Discussion' is the weakest point of the manuscript. In the current version, it only duplicates the methods and results, so that discussion as such is almost lacking. I would expect here a comparison of the new system with other methods that have been used to simulate herbivory (listed in lines 54-55), with careful discussion of problems and benefits of all these methods. Another interesting point is to discuss the differences in relative amounts of individual volatile compounds induced by different methods – to which biosynthetic pathways these compounds belong and what is their (presumed) function in plant defences against herbivory? It would be also important to know whether the SpitWorm only exists in the lab of their creators, or there is a plan to make it (commercially?) available to other groups of researchers. Replication of a device is critical to justify the validity of the results.
We have shortened the discussion but some values and methods from the Materials & Methods as well as from the Results section are necessary to explain why we did the different experiments, how they are connected and how they lead to our conclusions.
The presented study is a “proof of concept”. Experiments and discussions about the influence on biosynthetic pathways and the function of single compounds are beyond the scope of this work. Our aim is to present a method/device which can be used to answer these questions in the future.
To date the MecWorm/SpitWorm device exists only in our lab. We would happily provide the engineering drawings and computer program to any scientist who is interested in such a device. Please, contact the corresponding author. At the moment there is no plan to make it commercially available.
Introduction is generally fine, although the justification of the need to control both mechanical wounding and OS application can be strengthened by adding references to meta-analyses which compared plant responses to natural and simulated herbivory.
To be honest, we did not have a meta-analysis in mind when writing the paper… However, we added (according to reviewer’s 3 suggestion) a very recent review of Waterman et al. (Simulated Herbivory: The Key to Disentangling Plant Defence Responses. Trends Ecol Evol 2019, 34, 447-458) which emphasized the need for methods which can separate mechanical wounding from chemical elicitors in a way very closely to “true herbivory” and might be helpful.
Also the goal of the study is missing in the introduction; it appears only in lines 237-239.
We think the last paragraph of the introduction contains the goal of the study. To make it more clear we changed it to: ”Although the development of MecWorm provided deeper insight in our understanding of insect herbivory, it was necessary to take the next step in order to mimic insect feeding as close as possible. Thus, here an insect feedingmimicking device was engineered and established that combines both mechanical wounding and the simultaneous application of chemical elicitors to allow the study of their different influence on the plant’s wounding response. This so-called SpitWorm was tested in comparison with both MecWorm and S. littoralis larvae feeding on induced defense responses in lima bean.”
The starting part of the results (lines 98-117), to my opinion, should be moved to methods.
According to reviewer’s 4 suggestion we added a sub-heading “SpitWorm System Setup”
The sentence «Results of … below» (lines 114-115) does not contain any information and can be deleted.
Done
At several occasions (lines 120-122, 126-129, 176-179) the authors report numerical values both in the text and on figures. This duplication should be removed.
We’d like to keep the numerical values for the following reasons: (I) because some of them are necessary in the calculation of the amount of oral secretion left by a larva per bite (chapter 2.8), (II) to make it easier for the reader compare values (see reviewer’s comment on Fig. 3 below).
Also Figure S5 is not needed, because it duplicates Table 1.
We followed the suggestion of reviewer 3 and exchanged Table 1 and Figure S5.
I am confused by the comparisons between wounding sizes of lima bean leaves resulting from larval feeding periods of different length (5 min to 17 hours; Figure 3a). I fully understand that the absolute values of the consumed leaf area are needed for calibrating SpitWorm; but for which purpose the authors compare between these areas? The only meaningful comparison between damage imposed during time periods of different length would be the wounding rate (mm2/hour).
Here we show the different wounding sizes for the time periods used in the different experiments. Larvae were starved before starting the experiments and they don’t feed continuously. Therefore, we have chosen different time periods to let the larvae feed on a leaf and after that we determined the average wounding size. If we would take e.g. only the area fed off after 17 hours we will calculate 0.42 cm2 for a 1 h time period which is less than half of the area we experimentally determined. For the 5 min value it’s even worse. Here it will result in a 10 times lower calculated area.
The statement in line 164 («insect's OS went deeper…») is not supported by the data.
This is an observation from Figure 5a in comparison to 5b.
Fig. 5 is not really impressive, especially the panels (c) and (d), and difficult to understand and interpret. I suggest moving it to Supplementary materials.
We’d like to keep this figure. See response to reviewer 3
The biting rate (line 186) is reported as an interval (minimum-maximum?), whereas all other data are reported as mean + SD. Was this measurement replicated?
As described in the Methods section (line 390, original manuscript), the biting rate was determined using slow-motion video analysis. We observed biting rates between 3 and 5 bites/s. These three different values and not the average were taken separately for the calculation of the amount of OS left per bite.
Table 1 requires technical editing. The sign +/- before SD in the caption line should be removed. P values should be rounded to include 2 digits for p=0.10-0.99 and 4 digits for p=0.0001-0.0999. The three last columns are not needed, because actual p values are reported in the previous columns. If the authors wish, they may use e.g. boldface to highlight significant p values. Sample sizes (now shown at the bottom of the table) shall go to a caption; the numbers of components that show significant differences between treatments shall go to the text.
According to the suggestion of the editor and reviewer 3 the table was moved to Supplementary Materials and replaced by Figure S5 in the manuscript. We rounded all pvalues to 4 digits; a differential rounding (2/4 digits) looks confusing. We removed the last three columns with the indications of the significance levels and set p-values < 0.05 in boldface.
Numbers of components which demonstrate the differences between treatments from Table 1 (now Table S1) have been reported in the text (see L212ff original manuscript)
Materials and methods. I missed the following information: (1) Was the damage applied to one leaf per plant or to several leaves? (2) Were leaf size and leaf position controlled? (3) How the mechanical damage was applied – was it just punching by a needle? What was a needle diameter, what was a rate of damage? How the leaf was fixed during the period of damage? What was the shape of damage? (hole size, holes per leaf, hole position, etc.) (4) Were the environmental conditions (air temperature, illumination, humidity etc.) standard during all the experiments? These values need to be reported (as it is done for plant growth and insect rearing conditions), because e.g. insect feeding rate naturally depends on temperature. (4) Was there only one specimen of SpitWorm, which means that the replicated damage was applied one after another, or there were several devices so that the replicated damage was applied simultaneously?
As already shown in Figure 2 and stated in 4.10, 4.12 and 4.13 (now 4.11, 4.13 and 4.14) one leaf of one plant was used. To make this even clearer and to address the other questions we added an additional subsection 4.2. See also reviewer 3.
As stated in 4.2 we used fully developed leaves. There is a slight natural variation in size of this leaves but for the whole study the wounding size was of interest.
As already described in the first paragraph of 2. (now 2.1) a hollow needle was used for punching. Now included additionally in section 4.2
The language of the manuscript is clear, but some expressions used by the authors (e.g. «annihilate their insect enemies», line 40; «insect feeding like defenses», line 198) seems odd for me.
“annihilate” changed to “defeat” changed to: “…to test its abilities to provoke insect feeding-like defenses.” According to reviewer’s 1 suggestion
The authors used relevant literature, although several points in the beginning of Introduction are poorly supported by references – the cited papers are too narrow to support the very general statements.
Here we disagree with the reviewer. And, in addition, no one of the three other reviewers raised this point.
Reviewer 3 Report
p.p1 {margin: 0.0px 0.0px 0.0px 0.0px; font: 12.0px Helvetica} p.p2 {margin: 0.0px 0.0px 0.0px 0.0px; font: 12.0px Helvetica; min-height: 14.0px} span.s1 {text-decoration: underline} span.Apple-tab-span {white-space:pre}Review of Plants ms 550830 by Li et al.
This paper presents the calibration of a new mechanical device (SpitWorm) that is able to deliver oral secretions and conduct foliar tissue wounding in a way that closely mimics feeding by larvae. Moreover, it elicits responses much more similar to those elicited by larvae compared to a previous version of the apparatus that mimicked the wounding (pattern and rate) without the delivery of oral secretions (MecWorm). Apart from offering an important technological advance for research on responses of plants to insect herbivores, it adds to our understanding of the importance of the amounts of oral secretions and rates at which they are delivered on the elicitation of particular responses to herbivory.
The following generally minor points should be addressed to improve clarity of the paper.
The discussion seems overly repetitive of some of the methods and results, and could be shortened.
L37 Provide the taxonomic delimitation meant by “higher plants”
L45 This sentence is not clear. All defence responses are triggered by HAMPS and the wounding trauma? What does the wounding trauma involve, i.e., what plant signals are elicited by which particular components of the trauma (membrane rupture, cell wall rupture? The following sentences seem to indicate that the wounding trauma is part of HAMPS, and therefore, it would not be “besides the wounding trauma…”
L68 Did the authors expect a diversity of signals equal to that of insect herbivores? What would be the logic of that?
L69 The last idea in this paragraph does not follow from the previous statement.
L78 It would help the reader to know how the delivery of larval saliva was prevented.
L100 “unit” rather than “until”?
L119 Not clear: wounding sizes of what?
I suggest replacing “leaf wounding size(s)” with “amount of leaf area damaged” throughout the paper. (This still needs to be defined at its initial appearance on the paper as the amount of leaf area removed or damaged by a caterpillar or mechanical device to improve clarity.
L120 Are these the areas damaged by the apparatus after the times indicated?
Fig. 3 Is too blurry. Do the units change between a) and b) ? Does the control of b) damage the same amount of area as the 5 min treatment of a) ?
L151 It is not clear how the authors got to estimate the optimal flow rate for SpitWorm from the data they obtained in sections 2.1 – 2.3.
L159 Before this section, authors need to explain how they obtained the oral secretions.
Fig. 5 Images are two dark. It would be helpful to explain in the legend what was expected in each image (e.g., no fluorescence on c and d, equal fluorescence between a and b ?)
L176 Why were those particular OS:dye dilutions chosen for the comparison with S. littoralis? Were more dilutions initially tested?
L190 Where was the OS left, at the surface or somewhere in the mesophyll ?
L220 Why was the expression of those particular genes tested?
Table 1 has too much information and makes it difficult to follow the comparisons on the corresponding explanations on the Results section. The supplementary figure S5 conveys the same message much more clearly. I suggest making figure S5 part of the actual paper, and leaving Table 1 for the supplementary material. This will help following the discussion on L 319-326.
Discussion
L246 Was the intention to have SpitWorm mimic larval feeding as closely as possible? If so, explain here how different the biting rate and leaf area damaged per bite were between SpitWorm and real larval feeding, and if possible, explain why it was not matched more closely?
L252 Why is it important that the dye have a maximum fluorescence emission between that of chlorophylls a and b?
L254-262 Seems unnecessarily repetitive with material in the Results section
L281 It is confusing to call the initial flow rate used “optimal” when this was later adjusted to approach more the delivery of OS by larvae by means of dilution. (see also L151)
L300 What do the authors mean by “long time delivery”?
L321 Emphasize that the main difference is in MecWorm inducing a greater response than larvae (or SpitWorm).
L331-340 This section of the discussion requires the reader to refer to the supplementary figure S6, where figure 8 is repeated. I suggest replacing Fig. 8 with Fig. S6. This will make
L339 Strictly, the statement can only refer to the set of dilutions tested. Also, for consistency throughout the text, use “1:10 dilution…” instead of “10 times diluted”.
The authors should explain whether either of the robotic systems (MecWorm or SpitWorm) can be used only on detached leaves, and how this detachment could change the responses compared to un-detached leaves, and for how long the physiology of a detached leaf would resemble that of an un-detached leaf.
L383 Why was image analysis not used to determine the amount of leaf area damaged by larvae ?
L393 Consider replacing “suffocated” with “euthanized by immersion“
L411 Is it “Corporation” rather than “Cooperation”?
L511 Consider replacing “The level…different group values was evaluated” to “Differences among group means for the various response variables were evaluated …”
References missing:
Baldwin 1990
Waterman et al 2019
Author Response
Reviewer 3 The following generally minor points should be addressed to improve clarity of the paper.
The discussion seems overly repetitive of some of the methods and results, and could be shortened.
See response to reviewer 2
L37 Provide the taxonomic delimitation meant by “higher plants”
changed to: vascular plants L45
This sentence is not clear. All defence responses are triggered by HAMPS and the wounding trauma? What does the wounding trauma involve, i.e., what plant signals are elicited by which particular components of the trauma (membrane rupture, cell wall rupture? The following sentences seem to indicate that the wounding trauma is part of HAMPS, and therefore, it would not be “besides the wounding trauma…”
We rephrased the part to: But herbivory is more than the simple removal of plant tissue.[4] Besides the wounding trauma, defense responses of plants to an herbivore attack are triggered by compounds released by the herbivore. These stimuli are classified into two categories: (i) chemical elicitors derived, for example, from herbivore oral secretions, oviposition fluids or environmental DNAs (eDNA) that were left behind by insects (herbivore-induced molecular pattern, HAMPs); and (ii) those that originate from the specific patterns of wounding, i.e. the mechanical damage and the resulting elicitors from plants like oligosaccharides and peptides. [5-8] This second category is also called damage associated molecular patterns (DAMPs). Only both aspects together are able to induce the full spectrum of plant herbivory defenses [9].
L68 Did the authors expect a diversity of signals equal to that of insect herbivores? What would be the logic of that?
We rephrased the sentences to: “Compared with the vast diversity of herbivores that attack plants, the known herbivore-derived elicitors are relatively few. However, the molecular mechanism of plant perception of these known elicitors needs further study”
L69 The last idea in this paragraph does not follow from the previous statement.
See response to previous comment
L78 It would help the reader to know how the delivery of larval saliva was prevented.
Added: “…feeding by cauterizing or surgically removing the larvae’s labial salivary glands.”
L100 “unit” rather than “until”?
changed to: “… up to a hole…”
L119 Not clear: wounding sizes of what? I suggest replacing “leaf wounding size(s)” with “amount of leaf area damaged” throughout the paper. (This still needs to be defined at its initial appearance on the paper as the amount of leaf area removed or damaged by a caterpillar or mechanical device to improve clarity.
We changed it to: “In order to adjust the wounding sizes (i.e. the amount of leaf area damaged or eaten) to be generated by MecWorm and SpitWorm, leaf wounding sizes of different larval feeding periods were measured.” And we kept the term wounding size throughout the manuscript.
L120 Are these the areas damaged by the apparatus after the times indicated?
Yes. See Materials & Methods 4.10, 4.12, and 4.13
Fig. 3 Is too blurry. Do the units change between a) and b) ?
Fig 3. is blurred due to a bad pdf compilation. Yes, the units change between Fig 3a (cm2) and 3b (mm2).
Does the control of b) damage the same amount of area as the 5 min treatment of a) ?
For this we will keep numerical values in the text additionally to the graphs (see reviewer’s comment above). 5 min treatment in 3a (0.30 ± 0.13 cm2) control treatment in 3b (29.98 ± 4.29 mm2)
L151 It is not clear how the authors got to estimate the optimal flow rate for SpitWorm from the data they obtained in sections 2.1 – 2.3.
There must be a misunderstanding. Section 2.4 describes the estimation of the optimized flow rate which is completely independent from the results in sections 2.1 - 2.3.
L159 Before this section, authors need to explain how they obtained the oral secretions.
Collection of OS is described in the Methods part 4.4.
Fig. 5 Images are two dark. It would be helpful to explain in the legend what was expected in each image (e.g., no fluorescence on c and d, equal fluorescence between a and b ?)
The images are not processed in any kind and the fluorescence is weak but the pictures show that there is only fluorescence when we use fluorescent dye with S. littoralis or SpitWorm, respectively. The conclusion from these observations is discussed in detail in the Discussion section line 263 ff (original manuscript). We think discussions/explanations/expectations should not be part of a figure caption.
L176 Why were those particular OS:dye dilutions chosen for the comparison with S. littoralis? Were more dilutions initially tested?
We had to start somewhere and realized that those dilutions were appropriate; not more dilutions were tested
L190 Where was the OS left, at the surface or somewhere in the mesophyll ?
According to Figure 5 we think both, but this is irrelevant for the experiment. As described in 4.9 (now 4.10), we extracted the whole leaf tissue for estimating the amount of OS left on an in the leaf.
L220 Why was the expression of those particular genes tested?
See line 327f (original manuscript)
Table 1 has too much information and makes it difficult to follow the comparisons on the corresponding explanations on the Results section. The supplementary figure S5 conveys the same message much more clearly. I suggest making figure S5 part of the actual paper, and leaving Table 1 for the supplementary material. This will help following the discussion on L 319-326.
We exchanged Table 1 and Figure S5 accordingly.
Discussion
L246 Was the intention to have SpitWorm mimic larval feeding as closely as possible? If so, explain here how different the biting rate and leaf area damaged per bite were between SpitWorm and real larval feeding, and if possible, explain why it was not matched more closely?
Differences were already mentioned and discussed in the Discussion section. See e.g. L277ff (original manuscript). See also response to reviewer 4.
L252 Why is it important that the dye have a maximum fluorescence emission between that of chlorophylls a and b?
In order to have low background. To make it more clear, we changed the sentence to: “Lucifer Yellow CH was chosen because (i) of its fluorescence emission maximum at 535 nm, fitting perfectly in the green gap between 490 to 620 nm of chlorophyll a and b (no background at the wavelength of interest), (ii) it is assumed to be non-toxic, (iii) of its high quantum yield, and (iv) it is highly dissociated at physiological pH levels”
L254-262 Seems unnecessarily repetitive with material in the Results section
We removed L254-260 (original manuscript) but we kept the remark on the long-time effect of Lucifer Yellow.
L281 It is confusing to call the initial flow rate used “optimal” when this was later adjusted to approach more the delivery of OS by larvae by means of dilution. (see also L151)
We changed to “optimized” now. We also changed “optimal flow rate” to “optimized flow rate” in the whole manuscript The optimization of the flow rate (i.e. finding the lowest hydromechanically possible continuous flow with this system) is described in section 2.4. To adjust the amount of larval OS delivered at this flow rate to the amount of OS left by a larva (we called it ‘effective’ OS), dilution of the original OS was necessary. This we explained this in detail in the Discussion section (L277ff original manuscript).
L300 What do the authors mean by “long time delivery”?
Changed to: in experiments which last several hours
L321 Emphasize that the main difference is in MecWorm inducing a greater response than larvae (or SpitWorm).
This is the statement of the following sentence (L322; original manuscript)
L331-340 This section of the discussion requires the reader to refer to the supplementary figure S6, where figure 8 is repeated. I suggest replacing Fig. 8 with Fig. S6. This will make
Replaced
L339 Strictly, the statement can only refer to the set of dilutions tested. Also, for consistency throughout the text, use “1:10 dilution…” instead of “10 times diluted”.
We rephrased that statement: “Additionally, the results showed that 1:10 dilution of OS is an appropriate dilution factor to add OS to SpitWorm to mimic S. littoralis feeding”
The authors should explain whether either of the robotic systems (MecWorm or SpitWorm) can be used only on detached leaves, and how this detachment could change the responses compared to un-detached leaves, and for how long the physiology of a detached leaf would resemble that of an un-detached leaf.
Both robotic systems as well as the larval feeding were used on non-detached leaves. See Figure 2. To clarify this we added the following sentence to 4.1 Plants: “For all plant treatments (S. littoralis, MecWorm, SpitWorm) one un-detached primary leaf of a seedling was inserted in the cubicle of the robotic device”.
L383 Why was image analysis not used to determine the amount of leaf area damaged by larvae ?
See answer to respective question of reviewer 4
L393 Consider replacing “suffocated” with “euthanized by immersion“
Changed accordingly
L411 Is it “Corporation” rather than “Cooperation”?
Corrected
L511 Consider replacing “The level…different group values was evaluated” to “Differences among group means for the various response variables were evaluated …”
Replaced as suggested
References missing: Baldwin 1990
We added: Baldwin, I.T. Herbivory Simulations in Ecological Research. Trends Ecol Evol 1990, 5, 91-93.
Waterman et al 2019
We added: Waterman, J.M. et al., Simulated Herbivory: The Key to Disentangling Plant Defence Responses. Trends Ecol Evol 2019, 34, 447-458
Thank you very much for pointing us to this very recent review which we missed during finalization of the originally submitted manuscript.
Reviewer 4 Report
This paper describes a robotic device, spit worm, that inflicts both mechanical wounding damage on plants and releases an oral secretion onto the leaf simultaneously. The authors describe how they tested the parameters of the machine including, bite rate, OS volumes, wounding amounts, timing, and flow rate. Once the parameters were optimized, the authors compared VOC emission and RT-PCR of JA-responsive genes with MecWorm, Spitworm, and the larval herbivore Spodoptera littoralis. Based on VOC PCA analysis and RT-PCR, specifically of a PAL gene, the authors determined that Spitworm has a similar effect on plant responses as does S. littoralis feeding.
Overall, the authors conducted sound scientific experiments in a logical order and have developed an interested tool for understanding the molecular mechanisms of insect feeding. However, there are a few major issues with the manuscript which I will outline below.
Can the authors put somewhere in the introduction why one would use a mechanical robot to study plant defenses rather than just use actual insects? This only has to be 1-2 sentences.
Figure 3 (and 7) are illegible in both print and digital format. Please make sure the font is made bigger and improve the image quality so readers can see all axes clearly. With figure 7, is it possible to change the R/G/B coloration to something that can be understood on a black and white print out?
There are some issues with clarifying the optimal flow rate. The authors determined that a flow rate of 10nL/sec with a 1:10 OS dilution was optimal for Spitworm, however, in the discussion (lines 246-248), it is mentioned that Spitworm's values are different than "real" S. littoralis feeding. The authors need to describe more in-depth why this is and what the implications for this for future experimentation are (if any).
The values that were used to determine Vb should be made clear throughout the text. For example, in each section, the authors describe the experiments that were done to determine the values for final Vb (ex. lines 121,129, 149, 195). Although these are listed in lines 186-192, these should made clear to the reader that they are going to be used as a variable in a later model. For example, line 121, it could be noted that 5 minutes was to be used as t (300 sec) in the model and then discuss (figure 3) why this time was chosen. This is somewhat done on line 136 when the authors indicate that an injection volume (Vi) of 5uL was used in subsequent experiments, but could also mention that "this was the value that was used in the model for determining OS left per bite". I had to go back several times in the previous sections to determine where the values for the equation came from. They are listed in the manuscript, but the reader has to search for them, when they could be stated very clearly in each section.
Figure 8 should be replaced with Supplemental Figure 6 so that all the PCR data can be seen in the main body of the manuscript. Just showing the PAL data is equivocal and does not show the reader some of the similarities of Spitworm to MecWorm and S. littoralis when measuring JA-gene induction. Also, please include what HK genes were used to get relative expression values in the caption.
The authors need to include what instar stage the larva were when the guts were dissected and volume was determined. It should also be mentioned that Spitworm is mimicking this larval stage only which may be cause significantly different plant responses than would earlier install stages.
Figure 3a (although the axes are extremely hard to read) suggests that after 5 minutes S. littoralis ate 0.3 cm2 of tissue, but in the injection study, after only 5 minutes, in the control, S. littoralis at 30 cm2 of tissue (Figure 3b). Is there an error in the caption or were insects allowed to feed for more than 5 minutes? What could cause this rapid increase in appetite? Am I not reading this correctly? Please clarify.
My final major concern with the paper, and this was mentioned in the discussion on lines 295-296, is that the authors did not assay for any active/known elicitors in the Spitworm OS after it was filtered with a Cellulose Acetate/nitrate 0.22 uM filter. Many other studies that use wounding + OS (eg. Baldwin lab) as a surrogate for actual insect feeding simply "paint" on pure OS or diluted pure OS after collection. To determine if Spitworm "works" just like S. littoralis, it is necessary to know that the chemical interaction at the machine/plant interface is the same. If compounds like volicitin or other Fatty Acid-AA conjugates, are filtered out then Spitworm does not "act" the same as "real" insect feeding, even though VOC profiles and the RT-PCR results of 4 JA genes (which are also affected by wounding) might be similar. The authors even suggest in lines 324-326 that the OSs are responsible for the attenuation of VOC levels because they "turn down the sound". One critical step that is missing in this study is to assay for known OS elicitors in the Spitworm's effluent (after filtering), boil the OS, or simply don't filter the OS and use a larger syringe needle so that viscosity caused by macromolecules in the OS is not an issue. Furthermore, this is counter-intuitive to tritrophic responses where insect elicitors are critical for massive VOC emissions to recruit parasitoid wasps. For example, in one of the coauthor's papers from 2005 (Mithofer et al. 2005), mechanical wounding resulted in lower VOC levels compared to insect feeding in several cases. If the general consensus is that OSs attenuate the VOC wounding response, please cite those studies.
Some minor/grammatical errors that can be addressed are found on the following lines:
Line 36: Delete "Standing at the beginning of the food chain".
Line 41: Overuse of embellished adjectives
Line 57 needs a reference
Line 60: "For a period of time". Can you be more specific? Dates?
Line 87: Change the word "real amount" to "actual amount"
Line 98-113 (Including Figure 1). Would this be better suited for "Methods" than results? Or, give it a subheading of 2.0 "Spitworm System Setup"
Sections 2.5 and 2.6. Rather than cutting out wounded tissue and/or paper to extract dye and determine wounding area, wouldn't using a digital image analysis software like Image J be more robust? This might help calculate "yellow" pixels for fluorescence and before/after photos showing amount of tissue removed after insect feeding. Printing, cutting out, and weighing paper to determine wound size seems like there would be many sources of error and add to inaccuracy.
Line 186: Delete this sentence and include something that indicates you used video analysis to determine biting rate. For example... "Using slow-motion video analysis, we observed a biting rate (BR) of 3-5 for S. littoralis feeding on bean plants".
Line 200: Add the word "and" after indices.
Table 1: Is there any way to order these in groups according to significance?
Lines 208-209: Delete the terms "on one hand and on the other hand"
Lines 303-305 are confusing and needs to be reworded.
Line 331: A new paragraph indentation is not needed
Author Response
Reviewer 4
Can the authors put somewhere in the introduction why one would use a mechanical robot to study plant defenses rather than just use actual insects? This only has to be 1-2 sentences.
Maybe the reviewer missed it or we understand the reviewer’s suggestion wrong. Starting with the second paragraph (line 53) of the introduction we describe the “history” of studies on the contributions of mechanical and chemical effects of herbivory on plant’s defense which consequently lead to the development of the SpitWorm. Additionally, we changed the first sentence of paragraph two of the introduction.
“To study the contributions of the two aspects (mechanical wounding and chemical elicitors) the insect’s feeding behavior has to be emulated and separated from the “insect’s chemistry”. Mechanical wounding of insect feeding was originally mimicked…”
Figure 3 (and 7) are illegible in both print and digital format. Please make sure the font is made bigger and improve the image quality so readers can see all axes clearly. With figure 7, is it possible to change the R/G/B coloration to something that can be understood on a black and white print out?
We apologize this. The quality of Figures 3 and 7 is ok in the original Word document and the submitted tiff-files. The blurring in the PDF file results from a bad compilation of the word document. The symbols for the different treatments are changed in Figure 7 (now Figure 8) to distinguish them in b&w print-outs.
There are some issues with clarifying the optimal flow rate. The authors determined that a flow rate of 10nL/sec with a 1:10 OS dilution was optimal for Spitworm, however, in the discussion (lines 246-248), it is mentioned that Spitworm's values are different than "real" S. littoralis feeding. The authors need to describe more in-depth why this is and what the implications for this for future experimentation are (if any).
We changed the sentence to: “It is worth to note that due to the mechanical restrictions of an artificial device the values for… different to real S. littoralis feeding (see below).” The values for real S. littoralis feeding have been calculated and discussed in L277ff and 290ff (original manuscript)
The values that were used to determine Vb should be made clear throughout the text. For example, in each section, the authors describe the experiments that were done to determine the values for final Vb (ex. lines 121,129, 149, 195). Although these are listed in lines 186-192, these should made clear to the reader that they are going to be used as a variable in a later model. For example, line 121, it could be noted that 5 minutes was to be used as t (300 sec) in the model and then discuss (figure 3) why this time was chosen. This is somewhat done on line 136 when the authors indicate that an injection volume (Vi) of 5uL was used in subsequent experiments, but could also mention that "this was the value that was used in the model for determining OS left per bite". I had to go back several times in the previous sections to determine where the values for the equation came from. They are listed in the manuscript, but the reader has to search for them, when they could be stated very clearly in each section.
We added the following sentence at the end of the introductory paragraphs of the Results section: “The values used in the model for calculating the volume of OS per bite (Vb) left by a larva at the wounding edge (see Error! Reference source not found.) are indicated in the respective sub-chapters”.
The different variables used in the model for determining Vb are now indicated by: “Due to a stronger and clearer fluorescence signal at the wounding edges (Figure S2) an injection volume (Vi; see Error! Reference source not found.) of 5 µL of Lucifer Yellow solution was used in subsequent experiments and together with a feeding time of 5 min (t = 300 sec) in the model for determining the volume of OS left per bite (Vb; see Error! Reference source not found.).”
“Measuring of dissected foreguts (n =5) resulted in an average foregut length l = 4.3 ± 0.8 mm; average width d = 3.8 ± 0.4 mm resulting in an average foregut volume (Vg) of 49 ± 17.3 µL, (Table S1) which was used in the model for calculating the amount of OS left per bite (Vb; see Error! Reference source not found.)”
“…resulted in a concentration range close to the average concentration of OS (mean ± SD, 8.46 ± 0.52 nL·mL-1) left at the wounding edges by labeled larvae (Cd; see Error! Reference source not found.)."
Figure 8 should be replaced with Supplemental Figure 6 so that all the PCR data can be seen in the main body of the manuscript. Just showing the PAL data is equivocal and does not show the reader some of the similarities of Spitworm to MecWorm and S. littoralis when measuring JA-gene induction. Also, please include what HK genes were used to get relative expression values in the caption.
Figure 8 (now Figure 9) was replaced by Figure S6 and the figure caption was changed to “Expression of four JA responsive genes (LOX3, PAL, PR2, and PR3). Lima beans treated for 1 h, 3 h and 9 h with MecWorm (MW), S. littoralis (SL), and SpitWorm. P. lunatus actin housekeeping gene (PACT1) served as normalizer. (SW; 10 times diluted OS, delivery speed of 10 nL·s-1); n = 3 for each treatment, log2 transformed, one-way ANOVA, post-hoc test: Fisher’s LSD, treatments with identical letters are not significantly different.”
The authors need to include what instar stage the larva were when the guts were dissected and volume was determined. It should also be mentioned that Spitworm is mimicking this larval stage only which may be cause significantly different plant responses than would earlier install stages.
The larval instar used for all experiments is stated in chapter 4.1; 3rd-5th instar. Moreover, it is obvious that we cannot make a general statement for any larval stages or other species or taxa. Consequently, we didn’t do that. We don’t think that this has to be explicitly stated.
Figure 3a (although the axes are extremely hard to read) suggests that after 5 minutes S. littoralis ate 0.3 cm2 of tissue, but in the injection study, after only 5 minutes, in the control, S. littoralis at 30 cm2 of tissue (Figure 3b). Is there an error in the caption or were insects allowed to feed for more than 5 minutes? What could cause this rapid increase in appetite? Am I not reading this correctly? Please clarify.
We are sorry for the blurred figure 3 in the pdf-file. The wounding sizes are given in Figure 3a in cm2 but in Figure 3b in mm2. But thanks to this question, we discovered typos in the paragraph above Figure 3. Now all values for the 5 min treatments are given correctly in mm2.
My final major concern with the paper, and this was mentioned in the discussion on lines 295296, is that the authors did not assay for any active/known elicitors in the Spitworm OS after it was filtered with a Cellulose Acetate/nitrate 0.22 uM filter. Many other studies that use
wounding + OS (eg. Baldwin lab) as a surrogate for actual insect feeding simply "paint" on pure OS or diluted pure OS after collection. To determine if Spitworm "works" just like S. littoralis, it is necessary to know that the chemical interaction at the machine/plant interface is the same. If compounds like volicitin or other Fatty Acid-AA conjugates, are filtered out then Spitworm does not "act" the same as "real" insect feeding, even though VOC profiles and the RT-PCR results of 4 JA genes (which are also affected by wounding) might be similar.
We mentioned that we can’t exclude that some active compounds might be removed by filtering (L296 f original manuscript). But especially the elicitors known up to now are too small to be filtered off.
The results presented in our work clearly show that with the described set-up it’s possible to evoke a plant defense response almost identical to “true herbivory”. If compound are missing due to the filtering process then the used OS is actually kind of purified. Moreover, it has been shown that the method suggested above (Baldwin lab) doesn’t elicit any volatile production in lima bean and is more far away from “real” insect feeding than our approach (T. Koch, 2001, PhD thesis, Jena). We added this to the first paragraph in the discussion.
The authors even suggest in lines 324-326 that the OSs are responsible for the attenuation of VOC levels because they "turn down the sound". One critical step that is missing in this study is to assay for known OS elicitors in the Spitworm's effluent (after filtering), boil the OS, or simply don't filter the OS and use a larger syringe needle so that viscosity caused by macromolecules in the OS is not an issue.
The intention of this study was not to purify and identify all putative elicitors that might be present in OS but to develop a system that is able to mimic herbivory as close as possible. The reviewer is right that it is necessary to do more research in the future in order to isolate and identify the active compounds in the spit; however that is far beyond the aim of study.
Furthermore, this is counter-intuitive to tritrophic responses where insect elicitors are critical for massive VOC emissions to recruit parasitoid wasps. For example, in one of the coauthor's papers from 2005 (Mithofer et al. 2005), mechanical wounding resulted in lower VOC levels compared to insect feeding in several cases. If the general consensus is that OSs attenuate the VOC wounding response, please cite those studies.
As discussed (L306ff original manuscript) relative amounts of volatiles cannot be directly compared between these two studies because the use of activated charcoal Bricchi 2010) resulted in some cases to degradation of compounds leading to a decrease of e.g. (E)-β-ocimene in an unpredictable amount.
We didn’t state that OS attenuates the VOC wounding response in general but concluded for the lima bean/S. littoralis system (see L324ff, original manuscript):
“This leads to the conclusion, that in this case, … but compounds in the OS reduce the emission to ‘turn down the sound’.”
Some minor/grammatical errors that can be addressed are found on the following lines: Line 36: Delete "Standing at the beginning of the food chain".
Actually, we’d like to keep this sentence. Nothing wrong.
Line 41: Overuse of embellished adjectives
Changed to: “…passionate and intense interests…”
Line 57 needs a reference
Mithöfer et al. 2005 added here
Line 60: "For a period of time". Can you be more specific? Dates?
changed to: “Before the introduction of Mecworm,…”
Line 87: Change the word "real amount" to "actual amount"
changed
Line 98-113 (Including Figure 1). Would this be better suited for "Methods" than results? Or, give it a subheading of 2.0 "Spitworm System Setup"
Thanks for the suggestion. We added the sub-heading “2.1. SpitWorm System Setup”
Sections 2.5 and 2.6. Rather than cutting out wounded tissue and/or paper to extract dye and determine wounding area, wouldn't using a digital image analysis software like Image J be more robust? This might help calculate "yellow" pixels for fluorescence and before/after photos showing amount of tissue removed after insect feeding. Printing, cutting out, and weighing paper to determine wound size seems like there would be many sources of error and add to inaccuracy.
Yes, determining of wound sizes by digital imaging would be an option but we describe how it was done in these experiments. Weighing paper cutouts for area determination is a method known as “cut & weigh”. According to Grob et al., Modern Practice of Gas Chromatography, 4th Ed. John Wiley & Sons, 2004 p. 429, the error is about 2-3 % for chromatograms.
Line 186: Delete this sentence and include something that indicates you used video analysis to determine biting rate. For example... "Using slow-motion video analysis, we observed a biting rate (BR) of 3-5 for S. littoralis feeding on bean plants".
This is already described, in the way the reviewer suggested, in the Methods section. See line 390 (original manuscript).
Line 200: Add the word "and" after indices.
done.
Table 1: Is there any way to order these in groups according to significance?
We ordered the compounds by retention index making the comparison with other publications easier.
Lines 208-209: Delete the terms "on one hand and on the other hand"
done
Lines 303-305 are confusing and needs to be reworded.
Changed to: “In order to evaluate to what extend SpitWorm can mimic herbivory, relative VOC amounts in the headspace of leaves as well as expression levels of four JA responsive genes in leaves upon larvae, MecWorm, and SpitWorm treatment were compared.”
Line 331: A new paragraph indentation is not needed
Paragraph indentation removed
Round 2
Reviewer 2 Report
I am rather disappointed by the author's responses to my comments. The authors made only superficial changes in their manuscript, whereas several points which I consider important were not accounted for. I believe that the manuscript requires further work to improve its impact on the readers.
Reviewer 4 Report
The authors have addressed most of my concerns. Although I still feel that there are flaws in the overall use and overall understanding of SpitWorm, they are reporting on a new tool and their manuscript does not overstate the implications.